# Automatically Grading Rey-Osterrieth Complex Figure Tests using Sketch Recognition

Category: Research

## ABSTRACT

The Rey-Osterrieth Complex Figure Test (ROCF) is among the most widely used neuropsychological examinations to analyze visual spatial constructional ability and memory skills, but grading the patient's sketched complex figure is subjective in nature and can be time consuming. With increasing demand for tools to help detect cognitive decline, there is a need to leverage sketch recognition research to assist in detecting fine details within an ROCF's inherently abstract figure. We present a series of recognition algorithms to detect all 18 official ROCF details using a top-down sub-shape recognition approach. This automated grader transforms a sketch into an undirected graph, identifies and isolates detail sub-shapes, and validates sub-shape neatness via a point-density matrix template matcher. Experimental results from hand-drawn ROCFs confirm that our approach can automatically grade ROCF Tests on the same 18-item sketch detail checklist used by neuropsychologists with marginal error margin.

**Index Terms:** Applied computing—Health Care information systems; Human-centered computing—Gestural input; Human-centered computing—Mobile devices

## 1 INTRODUCTION

### 1.1 Rey-Osterrieth Complex Figures

The Rey-Osterrieth Complex Figure Test (ROCF), developed by Rey [41] in 1941 and refined by Osterrieth [37] in 1944, is a neuropsychological test that evaluates several cognitive functions including visuospatial abilities, memory, attention, planning, working memory and executive functions [28, 46]. The ROCF is characterized as a complex cognitive task [45], and is known in the field of neuropsychology as a useful metric for the frontal lobe function [44]. A participant is asked to copy the figure into a piece of paper, then copy it again two more times from memory. The shape is specifically designed to be abstract so that participants cannot associate it with any common object or concept. A clinician then grades all three sketches on whether 18 separate sub-shapes (henceforth called "details") exist and, if they do, how neatly they were drawn. A clinician grants up to 2 points for each detail that totals to 36 points, with partial credit given to distorted or misplaced shapes. Points for overall neatness of individual details is subjective and is generally dependant on an expert's intuition, especially for shapes that exist but might be drawn poorly. This results in different ROCF graders potentially producing two different scores. The proliferation of digital sketch recognition techniques and a push to digitize clinical neuropsychological examinations motivated our creation of an automated ROCF that can grade itself on the existing grading scheme.

From a digital sketch recognition standpoint, automatically grading an ROCF is non-trivial due to the complexity of the figure and test conditions resulting in inherently fuzzy sketch data. No two completed sketches are drawn in the same order, and very frequently shapes are drawn using portions from other shapes [13]. Bottom-up approaches tend to classify shapes as soon as their constraints are met, but shapes in an ROCF may in fact be only part of a detail or may end up as a portion of an entirely different one. A top-down approach not only more closely resembles a human grading an ROCF, but it also simplifies the recognition process by not needing

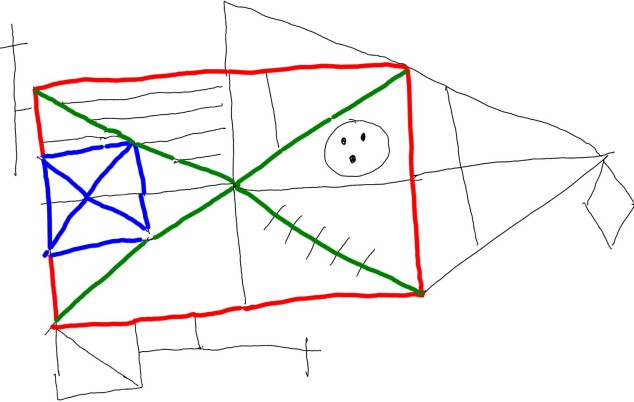

Figure 1: Our automated grader highlighting Details 2, 3, and 6 in red, green, and blue respectively.

to re-classify a shape at every step of the hierarchical recognition process.

### 1.2 Contribution

Significant research has been produced in analyzing the reliability of current rubrics [17, 32, 33, 48]. Automating the process started over two decades ago [13], but even recently surveys have cited a lack of contributions towards grading all 18 details at once. Moetsum et al. in research published in 2020 [34] specifies that "due to the unconstrained nature, these drawings, localization and segmentation of individual scoring sections become a highly challenging task" and existing work localizes only a "small subset of ROCF scoring sections".

Whereas previous efforts in automatically grading the ROCF can identify only a subset of the complex figure's details, we present the first fully automated ROCF grader that does not require user input to point to baseline shapes from which to begin recognition. Our contribution widely expands on Field's truss recognition technique field:2011:mechanix by introducing several graph traversal algorithms in order to isolate specific sub-shapes or regions from a given sketch. In addition to triangles, we also recognize squares, parallel lines, crosses, straight horizontal and vertical lines, and diamonds as well as shapes specific to the ROCF such as detail 6 (Cross with Square), detail 14 (Circle and 3 Dots), and 18 (Square with Line). Our system uses a multi-step recognition process that can identify shapes whether by crawling the resulting graph, by using template-matching shape recognition, or a combination of both, resulting in a more accurate and robust sub-shape recognition system for ROCF grading. Many of our recognition algorithms utilize well-known graph traversal and optimization algorithms (such as Dijkstra's Shortest Path [15] and Depth-First Search [47]). Our system represents the first fully-automated ROCF grader that recognizes the existence or absence of each of the 18 details and checks individual shapes for distortion.

To test our recognizer's performance, the system graded 141 digitized Rey-Osterrieth tests from participants and we compare

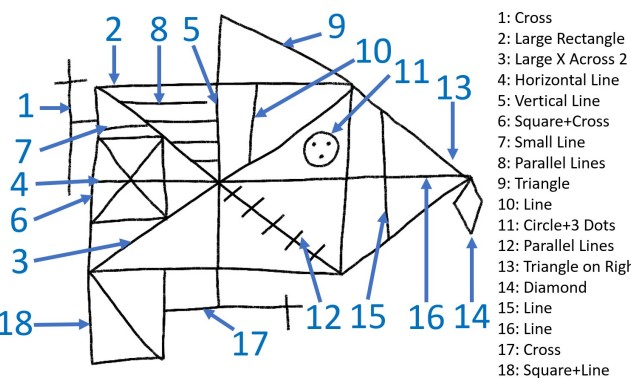

Figure 2: A Rey-Osterrieth Complex Figure Test, with all 18 Details listed.

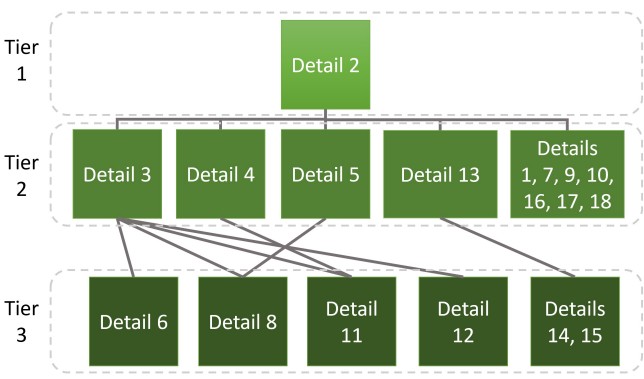

Figure 3: Auto ReyO's recognition hierarchy, designed to have as few dependencies as possible.

how closely our system's grades correlate with those of two expert graders. The experimental results demonstrates the proposed approach is successful in identifying the existence of sub-shapes within a large abstract shape.

## 2 RELATED WORK

### 2.1 Sketch Recognition Systems

Digital sketch recognition techniques favor bottom-up approaches that employ computational geometry to classify shapes [9, 11, 22, 27, 42]. Hierarchical sketch recognition systems such as LADDER [21], Sketchread [3], Chemink [38] and Mechanix [18, 18] generate composite figures by re-classifying shapes into more complex shapes in every step of the sketching process. In early bottom-up sketching approaches "steps" were typically separated by a UI button that explicitly instructed the system to create a recognition step. More modern systems, however, automatically separate "steps" by single-stroke actions and usually triggered when the user lifts their pen. This allows the system to continuously check the sketch to see whether the user is drawing a composite sketch made up of shape primitives.

Popular applications for bottom-up recognition of composite shapes using geometric primitives is especially popular in digital recognition of hand-drawn diagrams [1, 2, 6, 8, 10, 26, 53]. In these projects researchers seek to digitize hand-drawn flowchart and system design diagrams, interpreting diagram structure, flow of information, and preservation of variable and state checks through digital sketch recognition techniques [2, 26]. Circles, rectangles, diamonds, rhombi, and directional arrows [9] are used in diagrams to denote specific system or algorithm states or commands [1, 6]. Indeed, these projects originally served as the basis of *Auto ReyO*'s recognition due to the emphasis in recognizing primitives as part of a larger composite system of shapes. However, a chief difference between these projects and an ROCF sketch is that the ROCF by design has a large number of overlapping shapes, and specific details can be as granular as a single line within a specific area of other shapes. Diagrams and flowcharts, by contrast, are required to have clear spacing between its components and recognizing missing or distorted shapes is not a focus of these automated systems. While some form of composite figure recognition is necessary for automatically grading the ROCF, a top-down approach as explored in other systems [23] proved ultimately the most viable for *Auto ReyO*.

Corner detection also helps characterize digital shapes, with lightweight systems such as ShortStraw [55] and iStraw [56] being among the most efficient. *Auto ReyO* uses the open-source ShortStraw library in its recognition of corners and endpoints to generate the vertices during the graph creation stage. This is used in

tandem with line-intersection algorithms to segment the sketch lines such that individual shapes can be recognized. A frequent use case of this is recognizing details 4 and 6 of the ROCF (see Fig. 2). A user typically draws a single long line at once across the ROCF shape, so we are unable to use individual stroke order to recognize details, but rather need the segmentation that a line-intersection algorithm combined with ShortStraw is able to provide.

### 2.2 Template Matching Shape Classification Systems

The "Dollar" family of recognition systems [4, 5, 50, 51, 54] remains among the most well known single and multi-stroke gesture classification algorithms, and serve as the basis for our own template-matching recognition algorithm presented as part of our system. While most techniques rely on stroke order, geometric properties, and physical characteristics such as speed, acceleration, etc., the "$P+" recognizer calculates similarity via "point cloud" approximation [49]. A point cloud is generated by resampling both a template shape and an input shape on the same resampling parameters, overlaying the input shape on top of the template sketch matching its shape, centering, and orientation as close as possible, and iterating through every point finding the closest match between template points and input points. The distance between the points that are closest together are added cumulatively and are presented as the overall "distance" metric between the template shape and the input shape. The "$P+" recognizer returns the closest template match, identifying what kind of shape the user has drawn. This is especially flexible when the application in question necessitates recognition that is agnostic to stroke order. Our technique for shape recognition as described in Section 3.4 is based on the "$P+" recognizer, particularly the technique of calculating a "distance".

Our technique differs, however, in that rather than calculating distance via point-for-point comparison, we generate a fixed-resolution matrix of point density for both the template and the input shapes and calculate distance between cells of both matrices. This allows us to generate a more accurate grader for shape neatness. Indeed, "$P+" only focuses on finding the *closest match* to a template since it is a shape classifier, but its internal distance metric value does not perform well to gauge whether an input shape is poorly drawn next to its provided "ideal" template shape.

### 2.3 Hierarchical Sketch Recognition

Hierarchical sketch recognition approaches generally check drawn lines to see if they meet requirements for a composite shape [29, 31]. Layered hierarchical systems for graph creation have been applied to both bottom-up and top-down systems [23], and involve the decomposition of a drawn sketch to specific broad categories by analyzing

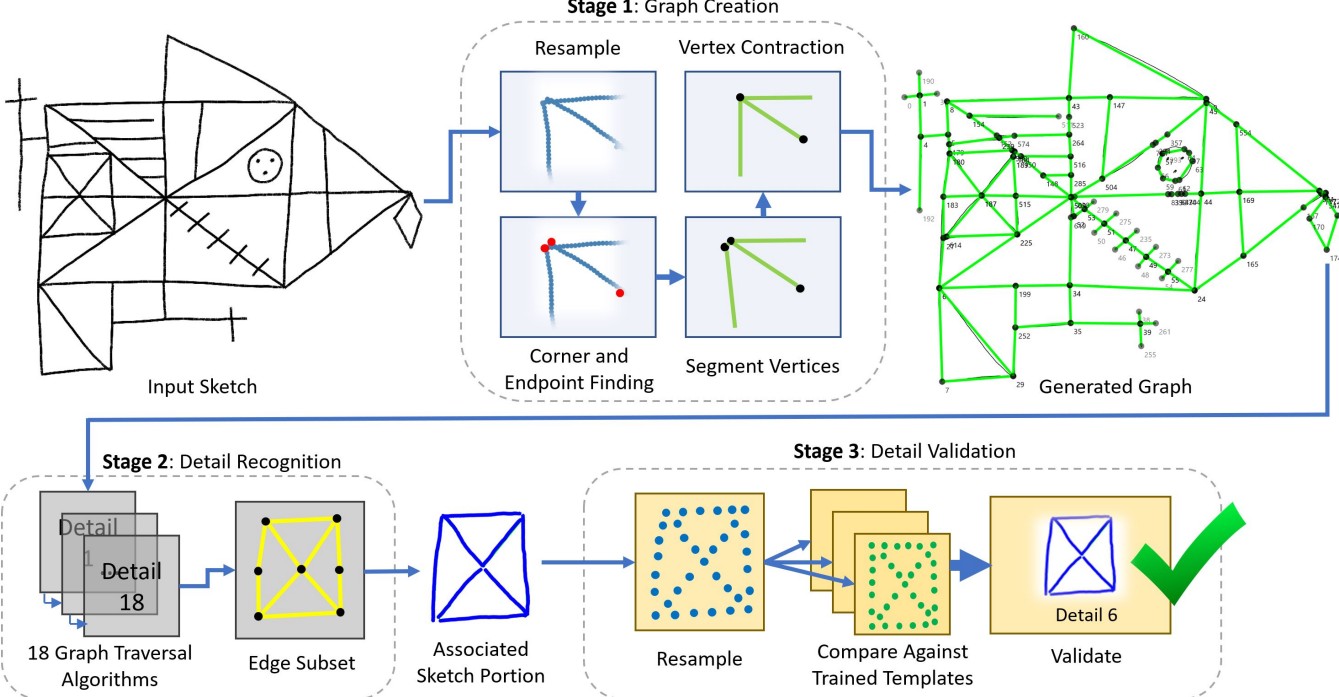

Figure 4: Description of ROCF sub-shape recognition system. Stages 2 and 3 shown in the figure are repeated for each of the 18 details of a Rey-Osterrieth complex figure.

sub-graphs [14, 30, 57]. This is typically used in the fields of computer vision to help decompose a system to primitive parts and represent them as a tiered graph. We envisioned a similar hierarchical tiered approach to the recognition of an ROCF due to the nature of the drawn details. To draw detail 10 in an ROCF, for example, the user needs to have drawn both details 2 and 3 to be able to connect the line properly (see Figure 2). Similarly, detail 14 requires the existence of detail 13 to receive full marks for both correct placement and shape neatness. Rather than represent the entirety of the sub-shape as a single vertex in a graph, however, we envisioned the vertices of a graph being represented by intersecting lines and endpoints, and applied the concepts behind sub-graph composite object recognition to identify the ROCF details themselves. The cited foundational work on graph implementations to supplement computer vision and object recognition informed our own approach to automatically grade ROCFs using a graph itself as the vehicle for tiered object recognition.

### 2.4 Efforts to Automate Neuropsychological Examination Analysis

Efforts to automate other neuropsychological tests has renewed interest in sketch sub-object detection [7, 16, 35, 36]. Object recognition ranges across various neuropsychological examinations including clocks [24, 25] and general handwriting tasks [20, 39]. However, whereas recognized objects for these tests tend to have heavily distinct characteristics, ROCF details are mostly composed of simple primitives that appear frequently. For example, detail 5 shown in Figure 2 is defined not only as any vertical line, but rather a specific vertical line within the sketch. Work presented by Prange *et al.* [40] cites Rey-Osterrieth figures as a motivating factor in the need to identify geometric shapes inside complex abstract figures. Existing attempts to automatically grade ROCFs are semi-automated or do not implement detection of all 18 details [12, 13]. The most recent attempt automates grading using a deep-learning neural network but

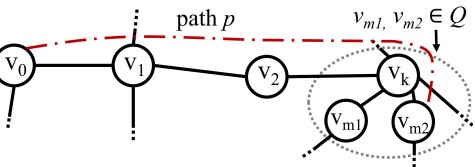

Figure 5: Finding path $p$ for the top horizontal side of detail 2's rectangle. Dotted area on right indicates *dist* radius. In this example $v_{m2} = c_1$, $dir = Right$ and $nextdir = Down$ (See Algorithms 1 and 2)

leaves ample room for improvement of individual segment detection, most notably single-line details [52]. Additionally, our system is able to produce a recognizer from only five training sketches to serve as templates, whereas neural networks require exponentially higher amounts of training data to function properly.

### 3 AUTOMATED REY-OSTERRIETH COMPLEX FIGURE TEST GRADER (AUTO REY-O)

*Auto Rey-O* is an application written on the Universal Windows Platform (UWP) that connects to a Neo SmartPen device via bluetooth for data collection. The same app is used to perform the fully automated grading process. *Auto Rey-O*'s top-down sub-shape recognizer divides the ROCF grading process into three distinct stages as shown in Figure 4.

### 3.1 Recognizer Generalizability

An important consideration of novel recognition and automation techniques in sketch recognition lies in articulating the generalizability and defining the constraints under which a presented technique aims to perform well.

Automating the ROCF motivates a brief discussion on generalizability due to the inherently "hard-coded" nature of its automation. Indeed, the complexity of the ROCF shape coupled with the requirements of detecting very specific lines necessitates a certain specificity of location and shape composition requirements. Some details, for example, are a single horizontal or vertical line, but of most importance is the location of the line relative to other details and the starting and stopping points. It is, in fact, this specificity in requirements that allows our method to recognize all 18 details, opposing previous work that only detects a subset of them.

At the same time, however, generalizability was taken into account when designing the recognizers that will be described in the following section. Generalizability was considered for two primary reasons. Firstly, our algorithm must be generalizable to recognize details despite a varying list of imperfections including but not limited to: crooked lines, shapes not entirely closed, various lines intersecting at different points, sharp angles accidentally being curved, the same line being drawn over several times, etc. The algorithm must also be able to, within reason, identify as many shapes as possible even in the absence of other shapes. Unless the shapes are directly dependent on each other for recognition, the absence or heavy distortion of one unrelated detail should not prevent the recognition of the other.

Secondly, as many recognition techniques as possible should be easily adaptable for other complex figure tests. As per the Compendium of Neuropsychological Examinations [46], seven complex figures are recognized as valid and tested figures for this purpose, and the Rey-Osterrieth Complex Figure test is the most popular. New variants with small changes are uncommon. The remaining six figures are: Taylor Alternate Version, Modified Taylor Complex Figure, and four Medical College of Georgia Complex Figures. All have similar size, complexity, and are a combination of straight lines, triangles, and simple geometric shapes. All contain a "detail 2": a large rectangle that serves as an anchor for the rest of the shapes. Our system was designed to be adaptable to recognize the 18 details of the remaining six complex figure tests by applying variations on the pathfinding algorithms on Table 1. Our three-stage method detailed in Fig. 3 can be adapted for all six remaining complex figure tests, so that extent we consider this approach generalizable for other complex figure tests of this type. Location heuristics need to be tailored for each detail, since the rules themselves are inherently specific and unique to the ROCF. We believe our three-step approach can be usable for any hierarchical sketch recognition problem involving complex figures where multiple sub-shapes must be discretely recognized but may share any number of lines.

## 3.2 Stage 1: Graph Creation

The graph creation stage is divided into four distinct steps. First, we prepare the sketch for corner detection by resampling to a uniform interspace length $S$ as follows:

$$S = \frac{\sqrt{(x_m - x_n)^2 + (y_m - y_n)^2}}{c} \tag{1}$$

where $(x_m, y_m)$ is the lower-right corner of the sketch, $(x_n, y_n)$ is the upper-left corner of the sketch, and c is a constant $c = 40$.

The second step utilizes the corner-finding algorithm from Wolin [55] to identify any "corner" from drawn strokes. To detect line intersections, two straight-line segments are compared with the target segment $y_a = a_2 x + a_1$ checked for intersection against comparison segment $y_b = b_2 x + b_1$ with equation 2.

$$\frac{a_1 + b_1}{a_2 + b_2} \in \left( x_1 - \frac{(0.15l)^2}{1 + a_2^2}, x_n + \frac{(0.15l)^2}{1 + a_2^2} \right) \tag{2}$$

where $x_1$ and $x_n$ represent the x values of the less and greater

vertices of the target segment respectively with $l$ being its segment length.

The third step creates undirected graph $G$, where every vertex $v$ is a line endpoint, corner, or intersection, and every edge $e$ is a drawn line connecting each $v$. Each $v$ contains a point from the sketch, and each $e$ contains the sampled points that connect the two vertices.

The fourth and final step performs vertex contraction on the created graph. Each vertex is iterated over and checked for near vertices that fall below a predetermined distance threshold. If two vertices are joined, their respective sampled points $s_i$ including points from an edge that might fall between them are combined into a single vertex $v$ containing all the sampled points. The distance measure is determined through complete distance used in hierarchical clustering, taking the maximum of the set in equation 3.

$$\{ ||s_i - s_j||_2 \mid s_i \in v_1, s_j \in v_2 \} \tag{3}$$

This serves both to connect segmented or near vertices and to reduce the overall complexity of the graph by eliminating edges. Finally, the vertices are iterated over a second time, checking if near vertices fall below a distance threshold, where the distance is determined through taking the minimum of the set in equation 3 referred to as single distance used in hierarchical clustering. If the distance falls below a predetermined threshold, the points are then are linked through an edge.

---

**Algorithm 1** Detail 2: Largest Rectangle

**Input**: Sketch Graph's vertex adjacency list
**Output**: Largest rectangle vertices

1: **for all** node $n$ in Graph **do**
2:     corner $c_1$=SideAndCorner($n$,*Right*,*Down*)
3:     corner $c_2$=SideAndCorner($c_1$,*Down*,*Left*)
4:     corner $c_3$=SideAndCorner($c_2$,*Left*,*Up*)
5:     corner $c_4$=SideAndCorner($c_3$,*Up*,*Right*)
6:     **if** $n = c4$ **then**
7:         add all SideAndCorner $b$ sides to rectangle $q$
8:     **end if**
9: **end for**
10: **return** largest $q$

---

## 3.3 Stage 2: Detail Recognition

All 18 ROCF details are recognized by applying a graph traversal algorithm to identify a "shape" within the graph. Each detail has an associated algorithm that is called in the hierarchical order defined by Figure 3. Every algorithm is designed to accommodate inherent graph imperfections from both the graph creation stage and the participant's hand sketch. For example, if the algorithm checks for a horizontal edge, there we allow a slope between 0.3 and -0.3 since participants are not expected to produce a perfectly horizontal line.

Every algorithm was designed to strike a balance between leniency to accommodate the imperfect nature of a hand-drawn shape and precision to find the expected shape if it exists. We are unable to thoroughly explain every detail's graph traversal algorithm, but we have chosen to explain detail 2 as Algorithms 2 and 1 since it is the most sophisticated of our recognizers and best illustrates our graph traversal approach.

### 3.3.1 Recognizing Detail 2

detail 2's recognition algorithm defined in Algorithm 1 and 2 is a greedy graph traversal algorithm tasked with finding the large rectangle that serves as the anchor for all other shapes in the graph. Our pathfinder finds one rectangle side $b$ and the corner at its end $c$ at a time. For each side *dir* is the intended path direction, and *dirnext* is the next path direction once we find our corner.

We define $N$ as single agents where every agent $n \in N$ has a start location $s_n \in G$ and a goal location $g_n \in G$. The path $p$ of $n$ consists of one side $b$ of our rectangle where $g_n = c$. Path $p$ is of length $k$ that is a sequence of vertices $p = \{v_0, v_1, v_2, ..., v_k\}$ such that each consecutive vertex is either in a defined direction *dir* (up, down, left, right, or diagonals) or Eucledian distance $r < 25$. At the end of our sequence $v_k$ one of two **conditions** is true:

1. $v_k$ is connected to a vertex $v_x$ such that direction of $(v_k, v_x) =$ *dirnext*

2. $v_k$ is connected to *other* vertices $v_m$ such that $r$ of $(vk, vx) < 25$.

The conditions describe that we have either (1) found a corner characterized by the start of the next side of the rectangle, or (2) the end of our sequence consists of various vertices very close together. This creates a set of "dead-end" nodes $Q$. For condition 1, $v_k \in Q$ and $c = v_k$. For any $v_m$ that satisifes condition 2, $\{v_{m1}, v_{m2}, ..., v_{mn}\} \in Q$. We check all vertices in $Q$ and return the vertex $v_e$ that satisfies condition 1. The final sequence is $p = \{v_0, v_1, v_2, ..., v_e\}$, and $c = v_e$. This is repeated four times to find the four sides of our rectangle, and return the largest such rectangle as detail 2.

### 3.3.2 Examples of Other Details

The rest of our graph traversal algorithms can be divided into two distinct categories: graph-crawling algorithms that identify shapes from the graph itself, and algorithms that use vertices as boundaries and then isolates all pen strokes within a specified region.

---

**Algorithm 2** SideAndCorner

**Input**: Start node $n$, directions *dir*, *dirnext*
**Output**: Side $d$ of rectangle, corner $c$

1: push $n$ to stack $s$
2: **while** $s$ not empty **do**
3:    **pop** $s$ to $p$, mark as visited
4:    **for all** adjacent pairs $(p_1, p_2)$ of $p$ **do**
5:       **if** direction of $(p_1, p_2) = dir$ **or** distance $e$ $(p_1, p_2) < 25$ **then**
6:          **push** $p_2$ to $s$
7:          $p = p_2$, repeat from line 5
8:       **end if**
9:       **if** dead end $p_n$ reached **then**
10:          **add** shortest path as a stack from $p$ to $p_n$ to set $c$
11:       **end if**
12:    **end for**
13: **end while**
14: **for all** current longest path $a$ in $c$ **do**
15:    **for all** adjacency pairs $(p_{a1}, p_{a2})$ of leaf $p_{an}$ in $a$ **do**
16:       **if** $p_{a2}$ is *dirnext* of $p_{a1}$ **then**
17:          **return** $c = p_{a1}$, $d = a$
18:       **else**
19:          **pop** $p_{an}$ from $a$
20:          **repeat** from line 15
21:       **end if**
22:    **end for**
23: **end for**

---

The former category is best for detail 2 as described previously, as well as simple shapes and lines like details 3, 4, 5, 7, 9, 10, 15, 16. The latter category is appropriate in cases when our graph generator may create highly variable graphs from imperfectly-drawn shapes, making it difficult for us to determine what the graph may look like. This is the case for details 1, 6, 11, 12, 14, and 17. In these instances we identify specific regions where we expect the detail to exist, and save all edges that are found. We isolate specific regions and run a bounded Depth-First-Search algorithm that returns all edges and vertices within the given region.

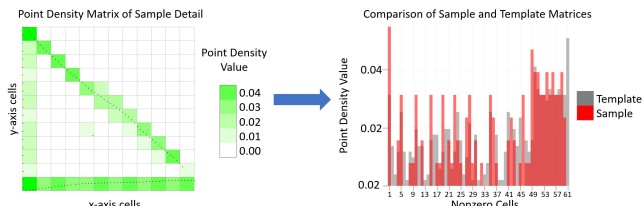

Figure 6: Comparing two point-density matrices to asses detail distortion: our sample, and a saved template. The process is repeated for all templates.

| Det. | Method of Recognition |
|------|----------------------|
| 1 | Isolate region, then DFS to fill |
| 2 | Greedy pathfinding, repeated per side |
| 3 | Dijkstra's between diagonal corners of #2 |
| 4 | Greedy single-direction pathfinding |
| 5 | Greedy single-direction pathfinding |
| 6 | Direction path for top/bottom, Dijkstra's |
| 7 | Greedy single-direction pathfinding |
| 8 | Connect horizontal lines bet. #3 and #5 |
| 9 | Find vertical, diagonal line above #2 |
| 10 | Greedy single-direction pathfinding |
| 11 | Isolate triangle, then DFS to fill |
| 12 | Isolate lower-right region, DFS |
| 13 | Find upward, downward diagonals |
| 14 | DFS to find all edges on tip of 13 |
| 15 | Greedy single-direction pathfinding |
| 16 | Greedy single-direction pathfinding |
| 17 | Isolate region, then DFS to fill |
| 18 | #2's technique, then single diagonal |

Table 1: General recognition method types for all 18 details. DFS is the Depth-First Search pathfinding algorithm. Dijkstra's is the Dijkstra Shortest-Path Algorithm

For detail 6, for example, our "region" is defined by the area inside the detail 2 rectangle and detail 3 cross, and we do not include the detail 4 horizontal line or detail 7 Small Segment in our DFS search. This returns the remaining subset of vertices and edges as seen in bottom portion of Figure 4. Table 1 describes the general method of recognition we applied to detect all 18 details in an ROCF, and the order of recognition is tiered as shown in Fig. 3.

The "region finding" category allows us to isolate some details, but if the shape is poorly drawn or missing entirely then isolating regions alone could not confirm shape neatness. This motivated the implementation of our third processing stage, which grades the isolated shape for correctness.

### 3.4 Stage 3: Detail Validation

Stage 3 compares only the **isolated sample** of the recognized detail to a set of template details to score the sample's quality. The system begins by centering, scaling, and resampling the isolated sample points so that they lie in a $[-1, 1]$ range on the $x$ and $y$ plane. We then create a map of some provided resolution $n$ such that a detail is represented as a $n \times n$ matrix where space in the figure is mapped to a cell of the matrix, each cell being some range $[x_i, x_j]$, $[y_i, y_j]$. Each cell is then given a p-value based on the number of points that lie within the range for each cell. A visualization of this is shown in figure 6. The same system is applied to each of the templates, then each template matrix is then averaged cell-wise to form a template mapping. The template matrix $Q$ is compared sampled detail matrix $P$ with equation 4 to determine how closely the two matched. The

| Det. # | $\Delta_{a,g1}$ | $\Delta_{a,g2}$ | $\Delta_{g1,g2}$ | F1-Score | Det. # | $\Delta_{a,g1}$ | $\Delta_{a,g2}$ | $\Delta_{g1,g2}$ | F1-Score |
|---|---|---|---|---|---|---|---|---|---|
| **1** | 0.69 | 0.68 | 0.37 | 0.787 | **10** | 0.21 | 0.20 | 0.10 | 0.962 |
| **2** | 0.18 | 0.44 | 0.33 | 0.857 | **11** | 0.47 | 0.45 | 0.21 | 0.878 |
| **3** | 0.07 | 0.11 | 0.16 | 0.978 | **12** | 0.46 | 0.44 | 0.11 | 0.872 |
| **4** | 0.27 | 0.30 | 0.12 | 0.927 | **13** | 0.13 | 0.11 | 0.11 | 0.966 |
| **5** | 0.08 | 0.49 | 0.49 | 0.978 | **14** | 0.42 | 0.41 | 0.15 | 0.881 |
| **6** | 0.31 | 0.56 | 0.26 | 0.958 | **15** | 0.50 | 0.49 | 0.09 | 0.770 |
| **7** | 0.47 | 0.26 | 0.13 | 0.855 | **16** | 0.21 | 0.19 | 0.08 | 0.919 |
| **8** | 0.28 | 0.33 | 0.13 | 0.966 | **17** | 0.57 | 0.66 | 0.34 | 0.788 |
| **9** | 0.35 | 0.20 | 0.07 | 0.904 | **18** | 0.45 | 0.49 | 0.21 | 0.925 |

Table 2: Classification results and average scoring differences for each detail across all graded tests. **n=141** for all details except for Detail 2, where **n=185**. $\Delta_{a,g1}$ denotes the **average point score difference** between Auto Rey-O and Grader 1, $\Delta_{a,g2}$ is the difference between Auto Rey-O and Grader 2, and $\Delta_{g1,g2}$ between Grader 1 and Grader 2.

best match is then found by shifting the the sample matrix by row and column to find the best possible position when compared to the template, given that some details will match in terms of their stroke and dimension but have somewhat different centers relative to the the template.

$$\sum_{i=1}^{n} \sum_{j=1}^{n} max\left(0, Q_{i,j} - P_{i,j}\right) \qquad (4)$$

The value of "distance" between template and sample is between 0 and 1, with 0 being the best. A shape that receives full credit for neatness is characterized as how close the sample is to the templates. Any value below 0.5 assigned to a sample is given full credit of 2 points. A value between 0.5 and 0.9 is given partial credit of 1 point. A value above 0.9 is given 0 points.

## 4 DATA COLLECTION AND RESULTS

We conducted a study with 68 cognitively healthy participants to complete a Rey-Osterrieth Complex Figure Test between the ages of 19-32. Although this test is meant to assess constructional ability and memory loss, healthy participants do not always score full marks on an ROCF [19], and indeed our testing corpus reflects a wide range of scores that conform to established normative data for our participants. All participants took the test in a simulated neuropsychologist's test environment and completed all three conditions (Copy, Recall, Delayed Recall). Participants were given a Neo SmartPen N2 and completed tests on pre-printed "blank" canvas pages that tracked the pen's location and instantaneously digitized all stroke data, allowing a more authentic testing experience since the ROCF is typically administered via pen and paper.

A total of 204 sketches from the 68 participants were collected. Of these, 5 perfect-score tests were set aside to be used as templates for Stage 3 validation. 14 were not gradable or their sketch data was corrupted, bringing the total graded to 185. All tests were also graded by two field experts whose grades we consider "ground truth" in this context. The first grader is a practicing clincial neuropsychologist and the second is a professor specializing in cognitive and visual perceptual rehabilitation in older adults. We measure our system's success in two ways: the F1-Score of our recognition algorithm for each detail, and the comparison between our system's total grade and the expert graders' total grade. For the latter, both our system's and the grades are on the 36-point scale as defined in Section 1.1.

A key factor considered when calculating F1-score was the subjectivity of distortion thresholds. While we implemented our own thresholds for distortion in Stage 3, instructions for the ROCF in the literature leave the definition of "distortion" at the discretion of the grader [46]. For recognition purposes we are interested in gauging whether our system can successfully either find a detail or confirm

its absence. The F1-score reports our system's ability to recognize the existence of a detail. Since we are still interested in comparing 36-point grades that also integrate distortion as partial credit, we also calculate Spearman's rank coefficient ($\rho = 0.767$) between our automatically-graded tests and those of our expert grader.

## 5 DISCUSSION AND LIMITATIONS

### 5.1 Results Discussion

In clinical neuropsychology, grading Rey Osterrieth Complex Figure tests has been the subject of constant iteration and is an active research topic, with numerous methods of interpretation being proposed and refined. As such, analyzing the process of grading these ROCFs automatically is not a trivial subject. Our analysis centered on simulating the perception of an detail since the granular differences in distortion are frequently attributed to grader subjectivity. Our aim, then, was the provide evidence the system perceived the details correctly, even if they might have been slightly distorted, or in the case of severe distortion the Stage 3 validation stage would be able to separate those clear cases. In terms of overall score comparisons, we sought to analyze how far apart individual test grades our system was from those of expert graders. Although the grades from individual graders were closer to each other than between each grader and *Auto ReyO*, our system compares favorably due to the high F1-score of the vast majority of details, and the average difference between *Auto ReyO* and our system being around 3 points out of 36 possible points for an ROCF test. We believe these results are significant in light of the fact that a fully automated ROCF grader that grades all 18 details has yet been proposed.

Also of note is the performance of detail 2, the large rectangle that serves as the anchor for the rest of the sketch. The organizational strategy score of the Rey-Osterrieth Complex Figure test places the highest priority on the existence of Detail 2 in a sketch due to its importance to the overall figure structure [43]. For the purposes of our system, this resulted in calculating F1-Score for recognition of the 18 details being conditional on whether detail 2 could be successfully recognized within a sketch. Exceptionally poor figures that lack a discernible detail 2 almost always result in very low or ungraded scores when hand-graded. Similarly, in very rare cases a poorly drawn ROCF could be graded by *Auto ReyO* if detail 2 could be recognized, while another ROCF that would score higher might not be graded due to a detail 2 that could not be recognized. For this reason, we have designed our recognition hierarchy such that the test is not graded if it cannot automatically recognize detail 2. This provides the most consistent application of grading requirements that is still consistent with the grading rubric as presented in the Compendium of Neuropsychological Examinations [46].

A total score of 0, however, is not necessarily due to a true negative. For 44 sketches, our algorithm was unable to find detail 2 due to a sloppy or unconnected drawing, but other details would exist. If we flatly calculated F1-Score of all Details for every sketch included

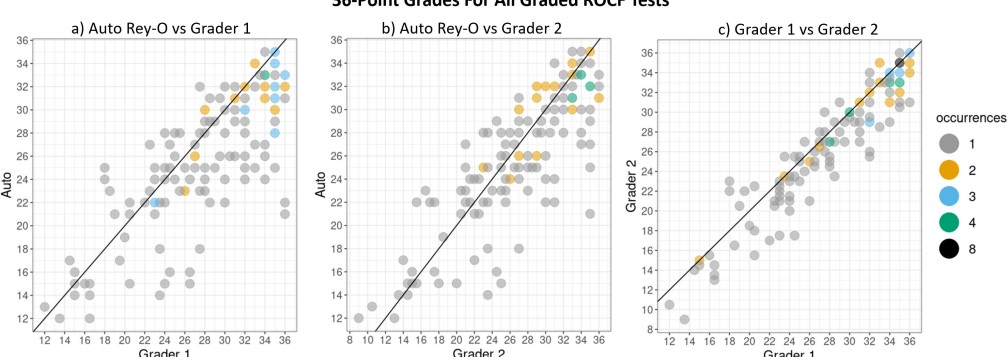

Figure 7: Grade plots for all 36-point scores, compared between Auto Rey-O and expert graders (**n=141**). (a) p=0.799, (b) p=0.829, (c) p=0.948

the ungraded ones, this would assign incorrect false negatives to the rest of the details. For that reason we tier F1-score calculations; for detail 2 we calculate it for all sketches (n=185), and for all other Details we calculate it where detail 2 was correctly detected (n=141).

The F1-Score results in Table 2 and the graph in Figure 7 demonstrate the effectiveness of Auto Rey-O. Our top-down system correctly identifies and validates the details with a high enough F1-score that shows the system working for typical test-takers. Table 2 also shows the average differences in scores (in points, maximum of 2) assigned between our Auto Rey-O system and our expert graders. All of the average differences in scores for each detail are well below 1 point and the vast majority below half a point, indicating a marginal difference in scoring between expert graders and our Auto Rey-O system. The system also works successfully for ROCFs with higher amounts of distortion. Such instances display the flexibility of the system in still identifying present details even if the participant has heavy lapses in memory.

The lowest-performing details are 1, 17 and 15. These had the highest amount of false negatives, although our manual review of these false negatives showed our system did recognize the details but chose to grant a score of 0 due to our threshold for distortion. Further refinements of distortion threshold values for these two details would improve their recognition quality.

Figure 7 compares the scores assigned by our automated grader and the two expert graders. Between the two expert graders, the correlation was $p = 0.948$, a Spearman's rank coefficient of $\rho = 0.942$, and an average difference in scores of $\Delta_{g_1,g_2} = 1.68$. Between our system and grader 1, $p = 0.799$, $\rho = 0.765$, and average $\Delta_{auto,g_1} = 3.21$. Between our system and grader 2, $p = 0.829$, $\rho = 0.802$, and average $\Delta_{auto,g_2} = 2.78$. Our automated system produced grades with a generally high correlation with those of the graders, although the grades from the experts were more similar to each other. In all three cases, low-scoring tests somewhat deviate across all graders, even between the expert graders. This is likely due to the aforementioned ambiguity in interpreting detail distortion. Our automated grader can also be observed to be consistently too strict on grading that produces consistently lower scores, which is partially attributed to the fact that it does not recognize details that were placed in the wrong location. In addition, at the suggestion of the expert graders who also served as domain experts, we chose to prioritize consistency in grading over leniency when deciding on partial credit thresholds since consistency is one of the key advantages of an automated recognition system.

### 5.2 Limitations

The main limitation of this graph-based approach to top-down sketch recognition is the reliance on line connections. Our vertex-

contraction algorithm in Step 1 of the system's process does connect lines with corners within a certain radius. We found this technique worked very well if sketches were drawn with reasonable neatness. If the lines are disconnected by more than half an inch, however, these lines will remain disconnected. This was a conscious design choice since vertex contraction cannot be too aggressive; otherwise, regions where any correct sketch would have high numbers of vertices would all get incorrectly contracted into one. This is the case such as the area where detail 6, 7, 3, and 8 all converge—even neatly-drawn sketches have a high concentration of vertices here. We intend to improve refine the recognition system to "jump" gaps and close disconnected lines only where appropriate.

### 6 FUTURE WORK AND CONCLUSION

Refinements can be made to help recognize specific kinds of poorly-drawn details. As previously mentioned, most sources of grading inaccuracies for our system came from poorly connected graphs due to sketch sloppiness. For healthy participants taking this test, our expert graders attributed sloppiness as a lack of effort rather than genuine memory loss if the patient has no hand motor issues. Still, there would be an interest in supplementing our graph traversal with connecting otherwise unconnected vertices to improve recognition performance.

Additionally, improvements to our Stage validation approach could be made to recognize finer details. Our validation method sometimes may not properly distinguish between small changes, such as an extra stray mark or one line missing. Identifying missing lines is important for details 8 and 12, where the number of parallel lines drawn is relevant to its grading. Our validation method is able to find these discrepancies somewhat frequently, but potential for improvement exists.

Lastly, we aim to work with clinical neuropsychologists to administer their test to willing clients to evaluate system usability in a clinical setting. This would produce additional sketch data taken from actual patients, and would allow us to perform UI/UX usability studies for clinicians. The ultimate aim of the system is to aid diagnosis process by automating the grading of an ROCF, so evaluating the user experience of clinicians as they collect the digital data and use the Auto Rey-O application for themselves is the next step to further this project.

Our Auto ReyO automatic Rey-Osterrieth Complex Figure test grader demonstrates the validity of a top-down sketch recognition approach using graph traversal algorithms. This significantly simplifies the recognition process where a bottom-up approach would need to take into consideration a prohibitively wide array of possible shape interpretations and re-interpretations. By employing graph crawling, classical vertex search, and optimization algorithms we

are able to identify key sub-shapes of geometric shapes.

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
