# OpenReview forum: "Automatically Grading Rey-Osterrieth Complex Figure Tests using Sketch Recognition"
_graphicsinterface.org/Graphics_Interface/2023/Conference_SD — Submitted to GI 2023 - second deadline_

### Official Review · Reviewer_fHr6 · 2023-04-17
**Interesting but how novel is this**

**Rating:** 4
**Confidence:** 4

**Review:**

This submission presents an algorithm that can be used to automatically score a Rey-Osterrieth Complex Figure test. The algorithm identifies 18 features (details) within a drawn figure using a combination of pathfinding / graph-based algorithms. The algorithm was then evaluated using figures that were drawn by healthy patients. The findings seem to suggest that the algorithm can detect most of the details at a level that is somewhat comparable to expert human raters. The contribution from this submission comes from the algorithm itself and the validation that shows which details it can (and cannot detail) well. This work does fit within the scope of GI.

I was not familiar with the Rey-Osterrieth Complex Figure test before reading the paper and am not an expert in sketch recognition. I have conducted research on pen-based and gesture recognition so I am familiar with stroke-based recognition methods.

Overall, this submission was easy to read and interesting, however, I have some concerns about the motivation, the novelty, and the lack of examples of the challenge in this task / what failures look like.

The submission does a good job explaining the challenges in automatically grading these types of figures, however, it is unclear how important developing an automatic approach to perform such grading is. How often are these tests administered? How long does it take an expert to grade a figure? What are the resulting grades used for? Once graded, would an algorithm provide a single score that would be used for something or does a doctor need to see specific examples of details that were poorly created? I struggled to answer these questions and thus understand how much of a contribution this algorithm would make.

Throughout the text, numerous sentences refer to how a complete automated grading solution has not yet been created, however, I was unclear as to which parts of this algorithm were new, different from other approaches, or the same as other approaches. As all of the detail recognition algorithms are not explained, it is difficult to read Section 3 / view Table 1 and determine what is novel here. Perhaps if Table 1 included references to prior work and delineated the current versus past approaches this would have made the novelty clearer.

Lastly, there aren’t any examples to illustrate what a good versus bad figure looks like (i.e., how difficult this problem is to solve algorithmically). Based on some of the text, there appears to be some fuzziness even with the expert ratings / interpretability. The description of the details is also unclear - Figure 2 doesn’t really explain how detail 10 differs from 15 or 16 (all noted as ‘line’).  On page 7 there is also a paragraph that explains how the recognition of details 1, 15, and 17 was poor, but there are not any examples to illustrate these failures – examples would explain how difficult this task is / how good the algorithms need to be, and thus speak to the (possible) success of this algorithm.

In summary, this submission is ok – it attempts to solve a niche problem, but as written, it is unclear what the novel aspects of the algorithm is and how much of a contribution this research will make to the field.

---

### Official Review · Reviewer_ZDEy · 2023-04-21
**Inadequate context about the research contribution in light of previous research**

**Rating:** 5
**Confidence:** 4

**Review:**

The paper describes a sketch recognition approach and system for a specific type of recognition problem - i.e., scoring drawings used in cognitive tests (ROCFs). The desire to improve the efficiency of diagnosis in these settings is worthy, but there are several limitations of the current submission that question its contribution to HCI or graphics.

First, the paper's review of previous literature is incomplete - in particular, it does not review all of the hierarchical sketch recognition techniques that could be legitimate competitors in the task of recognizing ROCF diagrams. For example:

Alvarado, C. (2011). Multi-domain Hierarchical Free-Sketch Recognition Using Graphical Models. Sketch-based Interfaces and Modeling, 19-54.

van de Panne, M., & Sharon, D. (2011). Flexible Parts-Based Sketch Recognition. Sketch-based Interfaces and Modeling, 153-179.

Deufemia, V., & Risi, M. (2020). Multi-domain recognition of hand-drawn diagrams using hierarchical parsing. Multimodal Technologies and Interaction, 4(3), 52.

Schneider, R.G.; Tuytelaars, T. Sketch Classification and Classification-Driven Analysis Using Fisher
Vectors. ACM Trans. Graph. 2014, 33, 1–9

Zhang, X.; Huang, Y.; Zou, Q.; Pei, Y.; Zhang, R.; Wang, S. A Hybrid convolutional neural network for
sketch recognition. Pattern Recognit. Lett. 2020, 130, 73–82.

Given the wealth of other work done in this area, the authors need to clearly state how their approach differs from these previous solutions - some of which clearly operate in similar spaces to that of the present work (e.g., recognition despite variation in subparts and optional subparts, see the van de Panne reference above).

In addition, the authors' claim about the lack of progress on the specific ROCF recognition does not provide adequate detail - e.g., the main recent paper cited here does not introduce the CNN-based solution proposed by Moetesum et al.

Second, the authors' evaluation of the tool is admirable in some respects but deficient in others. Comparing to expert human judges is a gold standard and a lofty goal - but as is reported in the paper, the algorithm does not come very close to matching human judgement, and so readers are left wondering whether the evaluation was successful or not. More information is needed about what are the implications for the level of accuracy found by the study - for example, it seems likely that the human experts would not accept the current algorithm's assessment for purposes of diagnosing actual patients. The paper states "Our automated system produced grades with a generally high correlation with those of the graders" but does not give any justification for why these correlations should be considered "generally high" and for what purpose. More information about where and why the algorithm made errors would help to contextualize the results and provide context for a possible contribution. For example, the paper states that "For 44 sketches, our algorithm was unable to find detail 2" - in how many drawings were the human experts unable to identify detail 2? Presumably none, so the authors should spend more time exploring this limitation (being unable to start recognition on a quarter of all drawings seems to be a major problem).

The authors also state in their evaluation that "these results are significant in light of the fact that a fully automated ROCF grader that grades all 18 details has yet been proposed." This statement side-steps the issue of whether other hierarchical recognition techniques could have done as well as the present algorithm if they had been appiled to the problem of recognizing ROCFs. Without a comparison to the current state of the art (both the systems mentioned by the authors and additional systems e.g. those above) it is impossible to determine whether the submission represents a contribution in terms of sketch recognition. The authors should clearly indicate what the most logical alternate system would be to theirs, and provide evidence that their system provides some advantage in this domain.

Third, there are a few issues that are not well explained or appear to be missing from the paper:
- the authors provide no definition of "F1-score". Assuming that they mean the F measure from information retrieval, using this measure without reporting recall and precision is a problem: reporting only score difference and F measure leaves several questions about how the approach worked: e.g., which figures had high false positive or false negative errors? did the algorithm make consistent mistakes e.g., failing to recognize a detail that was drawn in the wrong location?
- it is difficult to understand how the system's recognition hierarchy actually worked: e.g., how can detail 10 only depend on detail 2, and not detail 3?
- the system uses a digital pen for drawing the figure, but the authors do not explain why stroke order or timing is not used to improve recognition. The paper states that "No two completed sketches are drawn in the same order" but presumably there is substantial regularity that could be exploited in terms of order (e.g., some details would presumably be drawn as units). The paper also states that "A user typically draws a single long line at once across the ROCF shape, so we are unable to use individual stroke order to recognize details" but this appears to be a weak claim - e.g., most of the lines of detail 6 would be drawn together and after detail 2 (and not in "a single long line"); the lines of detail 8 would be drawn together and after details 5 and 3 (and not in a single long line). There may be cases where ordering cannot be used (e.g., interpreting scanned drawings) but in the use case presented here, the rationale for ignoring order and timing is not convincing.
- the authors also do not exploit the fact that a patient will draw three ROCFs; the authors should discuss why regularities on a per-patient level in the productions of these three drawings could not be used to improve accuracy.
- there are some typos and grammatical errors in the text: e.g., " the perception of an detail", "was the provide evidence the system", "so that extent we consider this approach generalizabled", etc.

Overall, the main concern is that the paper does not clearly indicate its research contribution in the context of previous work in sketch recognition, and does not provide evidence that the new algorithm improves on the performance of the state of the art.

---

### Official Review · Reviewer_T9xB · 2023-04-24
**Interesting application, questionable algorithmic choices, incomplete validation**

**Rating:** 6
**Confidence:** 3

**Review:**

The paper introduces a system that takes in a vector sketch and runs an automatic set of metrics to measure the user's cognitive abilities. Each test is based on recognition of some shapes and their relative positions and is a combination of corner detection, graph simplification, and some standard graph algorithms, such as traversals or shortest paths. They validate their system by comparing it to manual 'graders', showing the correlation.

As an application, I think this is a great area. I think these tasks clearly should be automated even if used just as a suggested score to a medical professional. The algorithm itself, even though not all the details are given (which is too bad, I would appreciate a complete description, e.g., in the appendix or supplementary materials). The algorithms themselves make sense, but I can't help wondering why the authors decided to develop custom, and often rather complex algorithms, instead of relying on some standard recognition systems. Why not, for instance, use primitive recognition from some system like Free2CAD, where they use deep learning to output primitives, to simplify the test later? If somehow this doesn't work and deep learning is not an option (although I don't see why, those are simple shapes), why not use some legacy computer vision descriptors (SIFT etc.) instead of this custom procedure in, e.g., Sec 3.4?

Assume there is a good justification for these choices, some algorithm design choices are still rather strange. For example, why first resample and then find corners? (also, what's "interspace" in the context of resampling?) Doesn't uniform sampling break the corners? I guess the authors are assuming the sampling is fine enough, but it's still strange, why is this even necessary at this stage? I would expect to first find corners, then resample preserving corners. Why first simplify graph then infer, e.g., Detail 11 that depend on the fact that it's a circle? When simplified, it must be harder to classify correctly?

Finally, I'm somewhat worried that all the user studies were done on healthy individuals, which makes me wonder whether the system will be robust enough to work on patients with some decrease in mental capacity, the very purpose of this system after all? For instance, will the corner detection be robust to shaking hands? How well will it match the manual grader for lower scores in general? Furthermore, I'm not convinced that just showing correlation is enough: for me, it seems that the maximum difference between the manual and automatic grades are more meaningful than just correlation. Luckily, this one is easy to fix from Fig.7.

Overall, I appreciate the system, but I'm having troubles understanding its algorithmic choices and am worried that it might not work in clinical setting.